# Preclinical Pharmacokinetics and Acute Toxicity in Rats of 5-{[(2E)-3-Bromo-3-carboxyprop-2-enoyl]amino}-2-hydroxybenzoic Acid: A Novel 5-Aminosalicylic Acid Derivative with Potent Anti-Inflammatory Activity

**DOI:** 10.3390/molecules26226801

**Published:** 2021-11-11

**Authors:** Mara Gutiérrez-Sánchez, Aurelio Romero-Castro, José Correa-Basurto, Martha Cecilia Rosales-Hernández, Itzia Irene Padilla-Martínez, Jessica Elena Mendieta-Wejebe

**Affiliations:** 1Laboratorio de Biofísica y Biocatálisis, Sección de Estudios de Posgrado e Investigación, Escuela Superior de Medicina, Instituto Politécnico Nacional, Plan de San Luis y Salvador Díaz Mirón S/N, Colonia Casco de Santo Tomas, Ciudad de Mexico 11340, Mexico; mgutierrezsa@ipn.mx (M.G.-S.); marcrh2002@yahoo.com.mx (M.C.R.-H.); 2División de Ciencias de la Salud, Universidad de Quintana Roo, Av. Erick Paolo Martínez S/N, esquina Av. 4 de marzo, Colonia Magisterial, Chetumal 77039, Mexico; 3Laboratorio de Diseño y Desarrollo de Nuevos Fármacos e Innovación Biotecnológica, Sección de Estudios de Posgrado e Investigación, Escuela Superior de Medicina, Instituto Politécnico Nacional, Plan de San Luis y Salvador Díaz Mirón S/N, Colonia Casco de Santo Tomas, Ciudad de Mexico 11340, Mexico; corrjose@gmail.com; 4Laboratorio de Química Supramolecular y Nanociencias, Departamento de Ciencias Básicas, Unidad Profesional Interdisciplinaria de Biotecnología, Instituto Politécnico Nacional, Av. Acueducto S/N, Colonia Barrio La Laguna Ticomán, Ciudad de Mexico 07340, Mexico; ipadillamar@ipn.mx

**Keywords:** 5-5-{[(2E)-3-bromo-3-carboxyprop-2-enoyl]amino}-2-hydroxybenzoic acid (**C1**), aminosalicylic acid, RP-HPLC, pharmacokinetics, ulcerative colitis, Crohn’s disease

## Abstract

Compound 5-{[(2E)-3-bromo-3-carboxyprop-2-enoyl]amino}-2-hydroxybenzoic acid (**C1**), a new 5-aminosalicylic acid (5-ASA) derivative, has proven to be an antioxidant in vitro and an anti-inflammatory agent in mice. The in vivo inhibition of myeloperoxidase was comparable to that of indomethacin. The aim of this study was to take another step in the preclinical evaluation of **C1** by examining acute toxicity with the up-and-down OECD method and pharmacokinetic profiles by administration of the compound to Wistar rats through intravenous (i.v.), oral (p.o.), and intraperitoneal (i.p.) routes. According to the Globally Harmonized System, **C1** belongs to categories 4 and 5 for the i.p. and p.o. routes, respectively. An RP-HPLC method for **C1** quantification in plasma was successfully validated. Regarding the pharmacokinetic profile, the elimination half-life was approximately 0.9 h with a clearance of 24 mL/min after i.v. administration of **C1** (50 mg/kg). After p.o. administration (50 mg/kg), the maximum plasma concentration was reached at 33 min, the oral bioavailability was about 77%, and the compound was amply distributed to all tissues evaluated. Therefore, **C1** administered p.o. in rats is suitable for reaching the colon where it can exert its effect, suggesting an important advantage over 5-ASA and indomethacin in treating ulcerative colitis and Crohn’s disease.

## 1. Introduction

Cardiovascular diseases, diabetes, rheumatoid arthritis, asthma, dermatitis, and inflammatory bowel disease are all related to inflammatory processes [1,2,3,4,5]. Intense research efforts have been dedicated to discovering effective anti-inflammatory therapies. Today, the most common treatment is with non-steroidal anti-inflammatory drugs (NSAIDs), characterized by their analgesic, antipyretic, and anti-inflammatory effects [6,7,8,9,10]. Unfortunately, significant side effects have been associated with NSAIDs, including gastric ulcers, bleeding, and perforation. These undesirable toxic effects have been the major limitation to their use [11,12,13,14,15].

For example, indomethacin is the drug of choice to treat osteoarthritis and rheumatoid arthritis. Its mechanism of action is the inhibition of the cyclooxynase-1 (COX-1) enzyme [16,17,18]. In 35–50% of patients. However, oral administration of indomethacin has been associated with systemic and local upper gastrointestinal side effects, such as erosions, ulcerative lesions, and petechial bleeding in the mucosa of the stomach [19,20,21]. Other studies show that the oral administration of indomethacin in rats and humans causes ulcerative lesions in the gastric mucosa stemming from the generation of reactive oxygen species and lipid peroxidation [22,23,24].

5-aminosalicyate (5-ASA) or its prodrugs (e.g., sulfasalazine, mesalazine, olsalazine, and balsalazide) have been employed as first-line medications to treat ulcerative colitis for maintenance or remission [25,26]. These drugs can trigger some adverse effects, including diarrhea, nausea, vomiting, headache, abdominal pain, fatigue, weakness, hepatic abnormalities, arthralgia, and myalgia [27,28,29].

Hence, it is necessary to discover new alternative drugs capable of inhibiting myeloperoxidase (MPO) and producing an anti-inflammatory effect without generating such severe adverse effects [30,31]. For this purpose, our research group has focused on the development of 5-ASA derivatives. After being designed and synthesized in a previous study, they were assessed in vitro and ex vivo [32,33]. The in vitro assays evidenced antioxidant properties when using the 2,2′-azino-bis(3-ethylbenzo thiazoline)-6-sulfonic acid (ABTS) and 2,2′-diphenyl-1-picrylhydrazyl (DPPH) methods. Particularly, the compound 5-{[(2E)-3-bromo-3-carboxyprop-2-enoyl]amino}-2-hydroxybenzoic acid (**C1**) (Figure 1), can reduce the production of the free radical DPPH about 90% in comparison to 5-ASA which generates a lower reduction of this radical, being about 85% at the same concentration (0.408 mM). Interestingly, **C1** also exhibited anti-inflammatory activity in a 12-O-tetradecanoylphorbol acetate (TPA)-induced mouse ear edema model. As an inhibitor of MPO, its effect proved to be comparable to that of indomethacin according to an analysis with the *o*-dianisidine method [32,33].

In the preclinical testing of a new drug candidate, the lack of in vivo activity can be attributed to inappropriate pharmacokinetic properties or toxicity (the formation of reactive metabolites) [34]. The aim of the current contribution was to take another step in the preclinical evaluation of **C1** by examining its acute toxicity and pharmacokinetic profile. Acute toxicity was explored with the up-and-down OECD method, while the pharmacokinetic profile was established by administering the compound to Wistar rats through the intravenous (i.v.), oral (p.o.), and intraperitoneal (i.p.) routes. After p.o. administration, the distribution of **C1** was determined in organs and tissues. The bound and unbound fraction of **C1** in rat plasma was quantified and the blood/plasma (BP) partition coefficient was calculated.

## 2. Results

### 2.1. Acute Toxicity of ***C1***

Median lethal dose (LD_50_) values in Wistar rats were >2000 mg/kg and >1098 mg/kg for p.o. and i.p. routes of administration, respectively. Animals did not show any signs of toxicity with the p.o. route. During the necropsy, moreover, no macroscopic changes were observed in the liver, small intestine, colon, heart, spleen, stomach, or kidneys. Therefore, according to the Globally Harmonized System (GHS) of Classification and Labeling of Chemical Products, **C1** is classified in the categories 5 for the p.o. route and 4 for the i.p. route, indicating a low risk of toxicity of this compound.

### 2.2. Validation of the Method for the Determination of ***C1***

The extraction of **C1** from plasma and homogenized tissues and organs was optimized to achieve reliable and consistent recovery. Accordingly, different conventional chemicals agents were tested (alone and in distinct combinations) for protein precipitation: acids, acetonitrile, methanol, chloroform, ethyl acetate, and dichloromethane. A two-step procedure was developed for liquid–liquid extraction from plasma, using 450 µL of acetonitrile and 450 µL of methanol for 100 µL of plasma. The resulting recovery of **C1** was excellent (>80%). In addition, a clean chromatogram was obtained for a blank plasma sample. The mobile phase composition for chromatography was optimized, finding a good peak shape with an isocratic elution of the mobile phase, which consisted of acetonitrile 50%, methanol 10% and water 40% (using a Zorbax SB C-18 column: 150 mm × 4.6 mm, 5 μm), at 1.0 mL/min and detection at 305 nm.

#### 2.2.1. Linearity

All plasma calibration curves for **C1** were linear for the concentration range of 0.1 to 100 µg/mL and showed a correlation coefficient (R) greater than 0.99 (Appendix A).

A typical linear regression equation for the calibration curve was y = 15927x − 8768.7 (R^2^ = 0.9996), where y represents the area of the peak of **C1** (arbitrary units) and x the concentration of **C1** (μg/mL) in the sample (Figure 2).

The lower limit of the quantitation (LLOQ) value was 0.1 µg/mL with a coefficient of variation (C.V.) of 3.4% (Appendix A). The limit of detection (LOD) was established at 0.5 µg/mL with the ICH Harmonized Tripartite Guideline for Validation of Analytical Procedures [35], finding a C.V. of 5.6% (Appendix A).

#### 2.2.2. Intra- and Inter-Day Accuracy and Precision

The values of intra- and inter-day accuracy and precision found at 0.1, 10, and 100 µg/mL (low, medium, and high concentrations) of **C1** are summarized in Table 1. The intra-day precision varied from 4.4% to 14.9%, and accuracy varied from 89.3% to 99.5%. The inter-day precision varied from 3.7% to 11.1%, and accuracy varied from 81.5% to 94.9%. These results were acceptable for the quantification of **C1** in the pharmacokinetic analysis.

#### 2.2.3. Recovery

The method was established for extracting **C1** from the samples, either plasma, homogenized tissues, or organs. Briefly, 100 μL of a sample was placed in an Eppendorf tube and 450 μL of acetonitrile was added. The mixture was vortexed for 3 min, followed by the addition of 450 μL of methanol and vortexing for another 3 min. The absolute percentage of recovery of **C1** from plasma with this procedure was assessed at 0.1, 10, and 100 µg/mL (Table 2).

#### 2.2.4. Selectivity

Using the **C1** extraction procedure and the RP-HPLC method described in the methodology section, typical chromatograms were obtained (Figure 3) for blank plasma (Figure 3a), **C1** (Figure 3b), the biological matrix spiked with ketamine, xylazine and heparin (Figure 3c), and an in vivo plasma sample from a rat administered **C1** (Figure 3d). According to the results, the assay was selective for **C1** quantification. The blank rat matrix (plasma, homogenized tissues, or organs) did not display a signal that interfered with the **C1** retention time. The drugs frequently used in experimental protocols of pharmacokinetic studies, such as ketamine, xylazine, and heparin, did not interfere with the **C1** peak in the chromatogram (Figure 3c).

#### 2.2.5. Stability

The stability of **C1** was examined at four concentrations (0.1, 0.5, 10, and 100 µg/mL) in triplicate (Table 3), finding it to be stable in rat plasma stored up to 48 h at 25 °C and up to 72 h at 4 °C. **C1** samples were stable before the three freeze–thaw cycles of the quality control samples.

### 2.3. Pharmacokinetic Profiles of ***C1***

Pharmacokinetic profiles of **C1** administered p.o. (50 mg/kg), i.p. (75 mg/kg), and i.v. (50 mg/kg) are illustrated in Figure 4a–c, respectively. The values of all the pharmacokinetic parameters (for each via of administration) were calculated with a non-compartmental analysis (Table 4).

After **C1** was given i.v., its plasma concentration declined rapidly from 5–45 min and then more slowly from 45–540 min, manifesting the behavior of a typical two-compartmental model (Figure 4c). The mean residence time (MRT) for **C1** was 1.7 ± 0.4 h, the elimination half-life (t_1/2e_) was 0.9 ± 0.2 h, and the volume of distribution (Vd) was 5.2 ± 0.9 L/kg. When **C1** was applied by the p.o. and i.p. route, the pharmacokinetic profiles also fit a two-compartmental model. After p.o. dosing, the rapid absorption of **C1** allowed it to reach a maximum plasma concentration (C_max_) of 2.8 ± 0.1 µg/mL in 33 min (Figure 4a), with a t_1/2e_ of 2.5 ± 0.6 h, and an absolute oral bioavailability of ~77%. After i.p. administration of **C1** at 75 mg/kg (Figure 4b), C_max_ was reached at 33 ± 5 min, with a t_1/2e_ of 1.4 ± 0.2 h.

The pharmacokinetic model of **C1** was established with Kinetica 5.0 software, finding that the plasma concentration declined bi-exponentially. The parameters calculated for this model were: α = 0.042 min^−1^, β = 0.040 min^−1^, K_12_ = 0.018 min^−1^, K_21_ = 0.019 min^−1^, the elimination half-life α (t_1/2α_) = 16 min, and the elimination half-life β (t_1/2β_) = 3 h.

#### 2.3.1. Distribution of **C1** in Tissues

Tissue distribution was assessed after p.o. administration of **C1** at 75 mg/kg (Figure 5). The concentration of the compound in tissues and organs was determined at 10, 30, 90, 180, and 360 min after its application, finding the maximum **C1** concentration and an ample distribution in all tissues and organs (brain, heart, liver, spleen, lung, kidney, stomach, small intestine, colon, testicles, and muscle) at 10 min, followed by a gradual decrease during 6 h. **C1** was able to cross the blood-brain barrier and hemato-testicular barrier. The highest concentration of **C1** was detected in the small intestine, suggesting that this organ is the absorption site of the compound.

#### 2.3.2. Plasma Protein Binding of **C1**

Protein binding results are summarized in Table 5. The unbound fraction of **C1** at concentrations of 1 to 20 μg/mL ranged from 93.4% to 96.3%, indicating the fraction of **C1** available to cross into tissues.

#### 2.3.3. Blood/Plasma Partitioning to Find the Blood/Plasma Ratio of **C1**

BP ratios for **C1** (Table 6) obtained experimentally ranged from 0.40 to 0.54. A BP ratio less than 1 indicates that the compounds are free in the plasmatic phase and are not inside the blood cells [30].

## 3. Discussion

Experimental studies of pharmacological activity and toxicity are essential in the process of drug discovery and development, especially when there is a lack of correlation in preclinical assays between the in vitro and in vivo pharmacological activity of a compound. The main causes for these differences are a low value in relation to the absorption rate, apparent distribution, half-life elimination (t_1/2e_), and/or bioavailability. Because of inappropriate preclinical pharmacokinetic properties, approximately 40% of tested compounds are rejected in phase I clinical trials in humans [36,37,38]. Another important reason for failure in drug development is the toxicity of the compound stemming from the formation of reactive metabolites [38]. Both pharmacokinetic properties and toxicity can be assessed in animals.

As an essential part of the pre-clinical studies of **C1**, therefore, its pharmacokinetics and acute toxicity were herein evaluated in rats, and then compared to the properties of 5-ASA and indomethacin. **C1**, a novel 5-ASA derivative, was previously synthesized in our laboratory and examined for anti-inflammatory activity with an ex vivo model of mouse ear edema. Its potent anti-inflammatory effect is comparable to that of indomethacin, the reference drug for the mouse model [32,33].

Indomethacin acts as an inhibitor of both COX1 and COX2 but is more specific for COX1. It is widely used in the clinic to relieve moderate to severe pain, tenderness, swelling, and stiffness caused by osteoarthritis, rheumatoid arthritis, and ankylosing spondylitis, and is also administered to treat pain in the shoulder stemming from bursitis and tendinitis [18]. 5-ASA is prescribed to patients with inflammatory bowel disease (IBD), acting as a prodrug or in a mesalazine formulation [25].

For **C1** to be proposed as a therapeutic alternative for indomethacin and 5-ASA, it is necessary to identify its pharmacological and toxicological effects, as well as its pharmacokinetic parameters and compare them to the approved aforementioned drugs to explore the advantages and disadvantages of each.

**C1** demonstrated a relatively low acute toxicity and did not show any signs of apparent toxicity in the animals for 14 days. Moreover, the necropsy revealed non-macroscopic changes in the following organs: liver, small intestine, colon, heart, spleen, stomach, and kidneys. For the p.o. route of administration in rats, the LD_50_ value for **C1** obtained in this work was >2000 mg/kg, while the values previously reported for 5-ASA and indomethacin are 2800 mg/kg [39] and 12 mg/kg [40], respectively. Based on these LD_50_ values, **C1** and 5-ASA can be assigned to categories 4 and 5 of the GHS classification, since both categories indicate a relatively low danger of toxicity, compared to indomethacin that, according to its LD_50_ value, it belongs to the categories 1 and 2 of the GHS classification, in which a high risk of toxicity is attributed, being considered a substance fatal if it is ingested. The main toxic effect of indomethacin (like other NSAIDs), demonstrated in sub-acute and chronic toxicity tests, is ulceration of the gastrointestinal tract [23]. Thus, **C1** has the advantage of representing a much lower risk of toxicity than indomethacin in rats, which is comparable to that of 5-ASA. These experimental results are consistent with the previous theoretical evaluation of **C1** carried out in Osiris Property Server Explorer, in which a low toxicity index for this compound was predicted [32].

In the preclinical phase, it is essential to guarantee that test compounds have appropriate drug-like properties, including their pharmacokinetic parameters, before proceeding to examine such properties in the clinical phase [36,37]. Hence, the pharmacokinetic profile of **C1** was determined in rats for p.o., i.p., and i.v. administration, which allowed for an evaluation of its absorption from the gastrointestinal tract and intraperitoneal tissues.

First, the pharmacokinetic parameters were calculated with a non-compartmental model, which is appropriate for new chemical entities without previous pharmacokinetic information [41]. According to all pharmacokinetic curves constructed, the absorption process was based on a first order kinetics [42]. When the pharmacokinetic curves were adjusted to a compartmental model with Kinetica 5.0 software. However, the behavior of **C1** fit the two-compartment model, and its plasma concentration decreased bi-exponentially, being dependent on the distribution process [43,44,45].

Parameters of half-life α and β were t_1/2α_ = 16 min and t_1/2β_ = 3 h. Consequently, the t_1/2_ value of **C1** elimination was proposed to be greater than that obtained by the non-compartmental model. According to the distribution analysis, the main organs for the absorption of **C1** are the stomach, small intestine, and spleen. The reason that the spleen is one of the main absorption sites for **C1** could be explained since this organ has particular structural characteristics that allow the molecules transported by the blood to be completely permeable by the splenic microcirculation [46,47]. Based on this important property, it has been suggested that a large part of the drugs that are excluded from most other organs, is due to their properties of load, size, etc., thus can be distributed within the splenic parenchyma [46]. The compound showed high oral bioavailability (77%). The apparent volume of distribution of **C1** (5.2 L/kg) indicates its ample distribution in all tissues and organs. This was corroborated by the distribution study, which revealed that **C1** crossed the hemato-encephalic barrier and was probably deposited in some tissues [45].

The comparison of the pharmacokinetics of **C1** and 5-ASA favored the former compound. Whereas 5-ASA acts as a prodrug or is included in specific formulations to reach the colonic mucosa [48,49], **C1** rapidly arrives to the colon after p.o. route of administration to produce a direct effect. Moreover, it is worth mentioning that the elimination half-life (t_1/2e_) of 5-ASA by this route is short, oscillating its value from 0.5 to 1.5 h, while **C1** has a half-life time of elimination (t_1/2e_) of longer than 2.5 h [29]. Therefore, despite **C1** is a derivative of 5-ASA, this compound could show a different saturable metabolism [50]. This effect of **C1** is of great importance so that in the future, when the studies of this compound have been expanded, an optimal and safe dosage program can be determined to obtain the desired therapeutic effect, or when be necessary maintain a stable concentration of this compound [51].

On the other hand, the oral bioavailability obtained is higher for **C1** than 5-ASA or indomethacin. About 90% of the latter reportedly binds to plasma proteins [52]. The unbound fraction in plasma was higher for **C1** than 5-ASA, thus leaving a greater percentage of the former compound available for absorption by tissues, making it more likely to reach the site of activity and exert a pharmacological effect [45,53].

In summary, the current results demonstrate that **C1** is not toxic to rats, and its pharmacokinetic properties are adequate for a drug candidate. Indeed, it shows some advantages over indomethacin and 5-ASA. Therefore, this compound is worthy of clinical evaluation as a potential treatment for inflammatory disorders such as Crohn’s disease and ulcerative colitis.

## 4. Material and Methods

### 4.1. Chemicals and Standards

Acetonitrile, methanol, and HPLC grade water were purchased from Tecsiquim (Ciudad de Mexico, México), while acetic acid (analytical grade), polysorbate 80 (Tween) and propylene glycol were acquired from Sigma–Aldrich (St. Louis, MO, USA). Heparin 1000 UI/mL and 0.9% sodium chloride were supplied by PISA (Hidalgo, Mexico). Anesthetics were obtained from Vétokinol (Lure Cedex, France), and ketamine (CLORKETAM^®^) and xylazine (PROCIN^®^) from PISA Agropecuaria, S. A. de C. V. (Hidalgo, Mexico), all three for veterinary use.

A batch of **C1** of 99.1% purity served as the reference standard. **C1** was synthesized in the Supramolecular Chemicals and Nanoscience Laboratory of the Unidad Profesional Interdisciplinaria de Biotecnología, Instituto Politécnico Nacional (IPN). The molecular structure was elucidated at the Nanoscience and Micro and Nanotechnology Center (IPN) with infrared (IR) spectroscopy, ^1^H and ^13^C nuclear magnetic resonance (NMR), and mass spectrometry (MS) [32,33].

### 4.2. The Formulation for Administering ***C1***

An intravenous solution was prepared by dissolving 100 mg of **C1** in 10 mL of a mixture of propylene glycol, polysorbate 80, and sodium chloride (0.9%) at a ratio of 20:5:75, (*v*/*v*/*v*), respectively. The solution was vortexed for 3 min and then sterilized by filtration before application in a nylon syringe filter of 0.22 µm at a final concentration of 10 mg/mL.

### 4.3. Animals

Male and female Wistar rats were obtained from the bioterium of the Escuela Superior de Medicina (IPN) and Harlan Laboratories (Mexico City, Mexico). They were kept in standard polypropylene cages under controlled temperature (22 ± 2 °C) and a 12 h light/dark cycle, and provided food (standard rat chow) and water ad libitum. For all experiments, rats were allowed to acclimatize to the lab conditions during one week prior to undergoing assays. For the pharmacokinetic studies, animals were randomly divided into three groups (*n* = 6, body weight 300 ± 20 g). Eight female rats were employed in the up-and-down acute toxicity procedure. The animals were fasted for 8 h before and 4 h after drug administration but always given free access to water.

### 4.4. Ethics Statement

The study was conducted in agreement with the guidelines of the Declaration of Helsinki and approved by the Institutional Review Board (Comité de Investigación para el Cuidado y Uso de Animales de Laboratorio of the Escuela Superior de Medicina, IPN, Mexico City, Mexico, approval number ESM.CICUAL-02/27-07-2015). It complies with the Mexican norm in this matter (NOM-062-ZOO-1999, Technical Specifications for the Production, Care and Use of Laboratory Animals, SAGARPA), as well as the “Guide for the Care and Use of Laboratory Animals” of the National Research Council and National Institutes of Health (NIH Publications No. 8023, revised 1978). In addition, all the animals were treated according to the considerations of the humane endpoint [54,55]. Following the collection of samples, the animals were sacrificed with an i.p. dose of 72 mg/kg of sodium pentobarbital to perform a necropsy.

### 4.5. Acute Toxicity of ***C1*** Determined by the Up-and-Down Method

The first group of rats (male and female) was administered 175 mg/kg of **C1** by the p.o. or i.p. routes. According to the Guideline 425 (Acute Oral Toxicity: Up-and-Down Procedure) of the Organization for Economic Cooperation and Development [56], if the animal survives for 48 h after the first dose, a second animal is administered a higher dose. If the first animal dies, the second animal is given a lower dose. A dose progression factor of 3.2 can be used. Since the animals that received the first dose (175 mg/kg) in this study did not die, the next groups were administered 550 mg/kg and 2000 mg/kg. The animals were monitored for 14 days post-dosing and then sacrificed. The LD_50_ values were calculated with AOTStatPgm [56].

### 4.6. RP-HPLC Validation

#### 4.6.1. Rat Plasma Sampling

Rat plasma was obtained by cardiac puncture, which was performed rapidly after anesthetizing the animals with sodium pentobarbital. Blood samples were collected in heparinized tubes. The plasma was separated by centrifugation at 8000 rpm and 4 °C for 15 min and then stored at −80 °C until needed.

#### 4.6.2. Instrumentation and Analytical Conditions

**C1** was quantified in a liquid chromatograph Agilent 1260 Infinity Series (Agilent Technologies, Palo Alto, CA, USA) equipped with a quaternary pump delivery system (G1311B), robotic autosampler (G1316A), column thermostat (G1316A), and a multi-wavelength UV detector (G1315C). The results were analyzed with OpenLab CDS EZChrom. Separation was performed on a Zorbax SB-Phenyl column (5 µm, 4.6 × 150 mm, Agilent Technologies, Palo Alto, CA, USA) and UV detection was carried out at 305 nm. The column temperature was maintained at 25 °C, and the injection volume was 10 μL. The mobile phase consisted of a mixture of (A) acetonitrile, (B) methanol, and (C) water at a ratio of 50% A, 10% B, and 40% C, at a flow rate of 1.0 mL/min. The total analysis time for each sample was 5.0 min when using an isocratic elution. It was not necessary to equilibrate between each injection.

#### 4.6.3. Preparation of Calibration Standards

Six stock solutions at 1000 μg/mL of **C1** were prepared by dissolving 10 mg in a mixture of acetonitrile-methanol (50:50, *v*/*v*), adjusting the volume in a volumetric flask of 10 mL and storing the final solution at 2–4 °C. The stock solutions were stable for three months. Calibration standards were prepared daily from the stock solution at concentrations of 0.1, 0.5, 1, 5, 10, 50, and 100 μg/mL, and calibration curves were constructed in triplicate by plotting known concentrations of the standard versus the detector response area (Appendix A).

#### 4.6.4. Quality Control Samples

A set of stock solutions were prepared independently as the quality control samples with plasma from rats without treatment, at three different concentration ranges in triplicate. Briefly, appropriate volumes of stock solution were spiked into 100 µL of blank plasma, and the resulting solutions were vortexed for 5 min. Appropriate volumes of acetonitrile-methanol (50:50) were added until reaching a final volume of 1 mL. After the quality control samples were centrifuged at 8000 rpm for 10 min, the supernatant was passed through a nylon syringe filter with a 0.45 µm pore size.

#### 4.6.5. Validation Parameters

The method was validated according to the FDA Industry Guidance for the Bioanalytical Method Validation and ICH Harmonized Tripartite Guideline, Validation of Analytical Procedures [35]. An examination was made of linearity, intra- and inter-day accuracy and precision, recovery, LLOQ and LOD, selectivity, and stability [57].

The validation of parameters for linearity was carried out with a nine-point calibration curve constructed by assessing the standards in the aforementioned concentration range. Subsequently, the known concentrations of the standard versus the detector response area were plotted and finally the linear regression model was fitted, and the coefficient of determination calculated (R^2^ ≥ 0.980) [53,58]. Intra- and inter-day accuracy and precision were established by evaluation of different quality control samples (*n* = 5) at high, medium, and low concentrations of **C1** (0.1, 10, and 100 µg/mL) on five consecutive days. The values for accuracy and precision were expressed in terms of relative error (RE) and relative standard deviation (RSD), respectively. The values of intra- and inter-day precision should not exceed 15% and accuracy should be within ±15% for the quality control samples. The extraction recovery percentage of **C1** in rat plasma and other tissues and organs (brain, heart, liver, spleen, lung, kidney, stomach, small intestine, colon, testicles, and muscle) was determined by comparing the mean of peak areas of the processed quality control samples with those of the corresponding standard solutions spiked into the matrix (plasma, tissues, and organs), which had the same final concentration as the pure standards. The recovery of **C1** was examined at 0.1, 10, and 100 µg/mL [53,58]. The LLOQ was found from the last point of the calibration curve, where the average recovery value was ±20% of the nominal value with a C.V. ≤ 20%. The LOD was quantified by the following equation: LOD = 3.3 σ/S, where σ represents the standard deviation of the intercepts of the regression lines and S represents the mean of the slopes of the calibration curves of the analyte [55]. To examine selectivity, blank samples (plasma, tissues, and organs) from six rats were analyzed at the LLOQ concentration, then compared with the corresponding plasma samples spiked with **C1** [57,59].

In addition, the potential interference of heparin, ketamine, and xylazine was investigated. Method specificity was assessed by the values of peak identity and purity of **C1**. A peak purity angle that is less than the peak threshold angle is an indication of spectral homogeneity or purity of **C1**. Stability of **C1** in plasma was evaluated in triplicate using the quality control samples at four concentrations. Samples were stored at 25 °C for 12, 24, and 48 h (short-term stability), and three freeze-thaw cycles at −20 to +25 °C for 12, 24, and 48 h. Moreover, post-preparative storage was tested by examining the ready-to-inject samples stored in refrigeration at 4 °C for 24, 48, and 72 h. The concentration of **C1** after each storage period was compared to the initial concentration, which was determined for freshly prepared samples [57,59].

### 4.7. Pharmacokinetic Studies of ***C1***

On the same day as **C1** administration, the rats (three groups, *n* = 6) were previously cannulated for compound administration by the p.o., i.p., and i.v. routes. A 24 G catheter was placed in the right lateral tail vein for blood collection. A first group of rats received a single dose of 50 mg/kg applied p.o. with a 20 G gavage needle. A dose of 75 mg/kg of **C1** was injected i.p. into the rats in the second group. A single dose of 50 mg/kg was given i.v. to the third group by means of another catheter previously placed in the left lateral tail vein. The doses used in this study were selected based on previous pharmacokinetic studies reported in rats for 5-ASA [60] or for derivatives of this compound [61]. Animals were provided with the standard diet 4 h after dosing. Approximately 250 μL of blood samples were collected from each rat and put in heparinized tubes at basal and 5, 10, 20, 30, 45, 60, 90, 120, 180, 240, 300, 360, 540, and 720 min post-dosing. The volume of blood removed was supplemented with an equal volume of 0.9% sodium chloride solution. The blood samples were centrifuged at 8000 rpm for 15 min to harvest 100 μL of plasma.

Plasma samples were processed immediately. Briefly, 450 µL of acetonitrile was added to 100 µL of plasma and vortexed for 5 min. Subsequently, 450 µL of methanol was added, and the sample was vortexed again, followed by centrifugation to precipitate the proteins. Finally, 10 µL of the supernatant were used for evaluation of the plasma concentration of **C1** by RP-HPLC with the UV detection method, previously validated for the pharmacokinetic and complementary assays herein performed.

The plasma concentration–time data were plotted. For each route (Appendix A: p.o., Appendix A: i.p. and Appendix A: i.v., respectively), a non-compartmental analysis was conducted using the statistical moment theory. Pharmacokinetic parameters of **C1** were calculated on Kinetica 5.0 software (Adept Scientific Ltd), including the maximum plasma concentration (C_max_), time to maximum plasma concentration (t_max_), elimination half-life (t_1/2e_), the area under the plasma concentration-time curve from time zero to the last measurable concentration (AUC_0-t_), the area under the plasma concentration-time curve from time zero to infinity (AUC_tot_), mean residence time (MRT), clearance (CL), and apparent volume of distribution (Vd).

#### 4.7.1. Distribution of **C1** in Tissues

Fifteen male Wistar rats were distributed in five groups (*n* = 3) and rats were orally administered **C1** at a single dose of 75 mg/kg. Subsequently, a group of animals was sacrificed (*n* = 3) at 10, 30, 90, 180, and 360 min. The whole brain, heart, liver, spleen, lung, kidneys, stomach, small intestine, colon, testicles, and muscle were rapidly extracted and thoroughly rinsed in ice-cold saline to eliminate blood and other content. All tissues and organs were weighed on an analytical balance and immediately processed. Each tissue sample was homogenized with saline solution in a 1:3 (*wt*/*v*) ratio. The preparation process for analysis was the same as described above for plasma samples [57] (Appendix A).

#### 4.7.2. Blood/Plasma Partitioning and Blood/Plasma Ratio of **C1**

Appropriate amounts of **C1** stock solution at 10 mg/mL were spiked into whole blood to obtain samples (prepared in triplicate) at a final concentration of 1, 5, and 10 and 20 μg/mL (in a final volume of 600 μL), to be incubated at 37 °C for 4 h. Plasma was harvested from blood samples by centrifuging at 8000 rpm for 15 min, and then the concentration of **C1** was established using the RP-HPLC method and interpolating the response area into a standard curve prepared with **C1** and blank plasma. The BP ratio was calculated by dividing 5 and 10 µg/mL, respectively, by the concentration found in plasma separated from blood samples. The concentration of **C1** in blood cells is assumed to be equal to its unbound concentration in plasma [38].

#### 4.7.3. Plasma Protein Binding Assay of **C1**

Different quantities of **C1** stock solution at 10 mg/mL were spiked into 600 µL of freshly obtained blank rat plasma to give a final concentration of 1, 5, 10, and 20 µg/mL. The resulting samples were incubated at 37 °C for 4 h. After incubation, an aliquot of 100 µL was removed to determine the total concentration, and another aliquot of 500 µL was transferred to a 10 kD cut-off ultrafiltration device (Millipore Corporation, Burlington, MA, USA). Subsequently, samples were centrifuged at 2000× *g* and 37 °C for 2 h. The free drug concentration of **C1** was measured by evaluating 100 µL of ultrafiltrate with RP-HPLC, interpolating the main areas of the samples into a standard curve constructed with known amounts of **C1**. The percentage of protein binding was calculated with the following formula (1) [57,62].
Protein binding ratio (%) = [(1 − (drug ultrafiltrate))/(total drug)] × 100(1)

## 5. Conclusions

The acute toxicity and pharmacokinetics of **C1** were evaluated in Wistar rats, finding appropriate drug-like pharmacokinetic properties. The compound has some advantages over 5-ASA and indomethacin with respect to pharmacokinetics, including greater bioavailability after p.o. administration. In addition, **C1** demonstrated a low risk of toxicity, in contrast to the higher toxicity of indomethacin. Hence, **C1** is a good candidate for clinical testing to treat inflammatory diseases such as ulcerative colitis.

## Figures and Tables

**Figure 1 molecules-26-06801-f001:**
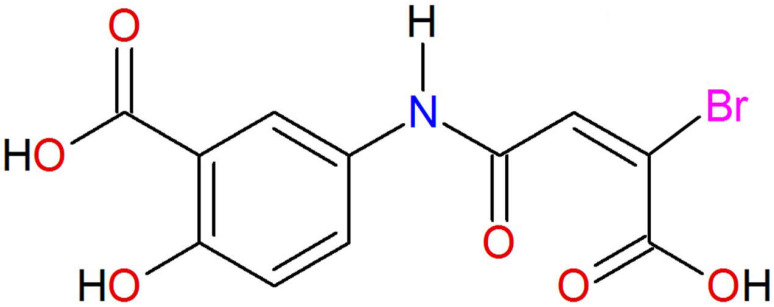
Chemical structure of 5-{[(2E)-3-bromo-3-carboxyprop-2-enoyl]amino}-2-hydroxybenzoic acid (**C1**).

**Figure 2 molecules-26-06801-f002:**
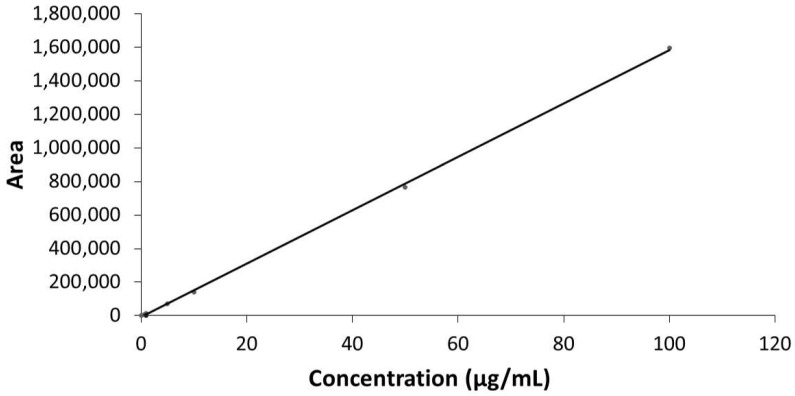
The calibration curve of **C1**. The calibration curves were constructed in triplicate, plotting the known concentrations of the calibration standards (0.1, 0.5, 1, 5, 10, 50, and 100 μg/mL) versus the response area of the detector.

**Figure 3 molecules-26-06801-f003:**
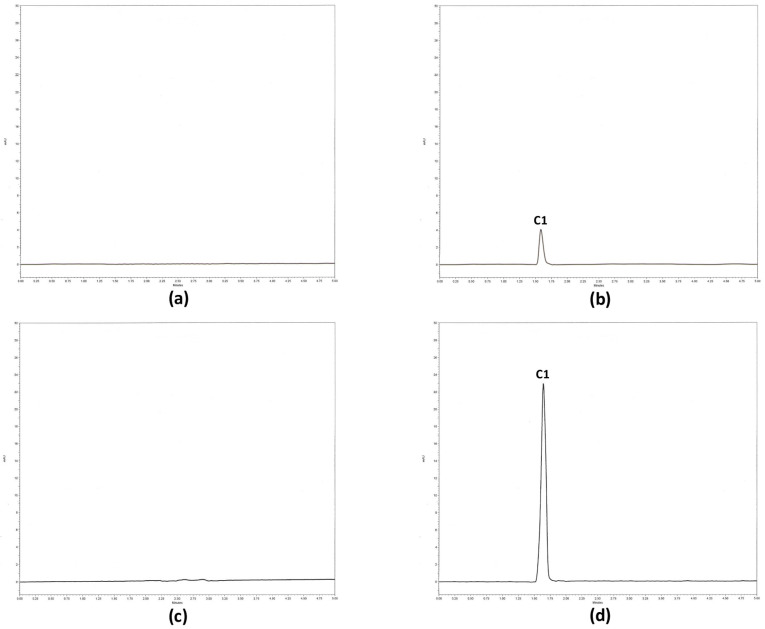
Selectivity chromatograms. The selectivity parameter was evaluated by the RP-HPLC method in the following samples: (**a**) biological matrix-plasma, (**b**) 1 µg/mL of **C1** as a standard, (**c**) a biological matrix enriched with ketamine, xylazine, and heparin, and (**d**) an in vivo plasma sample from a rat administered **C1**.

**Figure 4 molecules-26-06801-f004:**
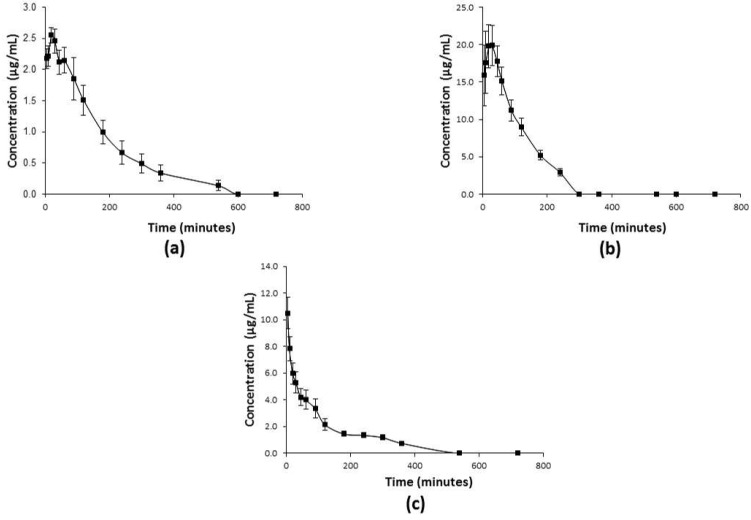
The plasma concentration-time curves for **C1** in rats (*n* = 6) found using a non-compartmental model, based on the mean ± SD value after administering the compound by the (**a**) p.o. (**b**) i.p., and (**c**) i.v. routes.

**Figure 5 molecules-26-06801-f005:**
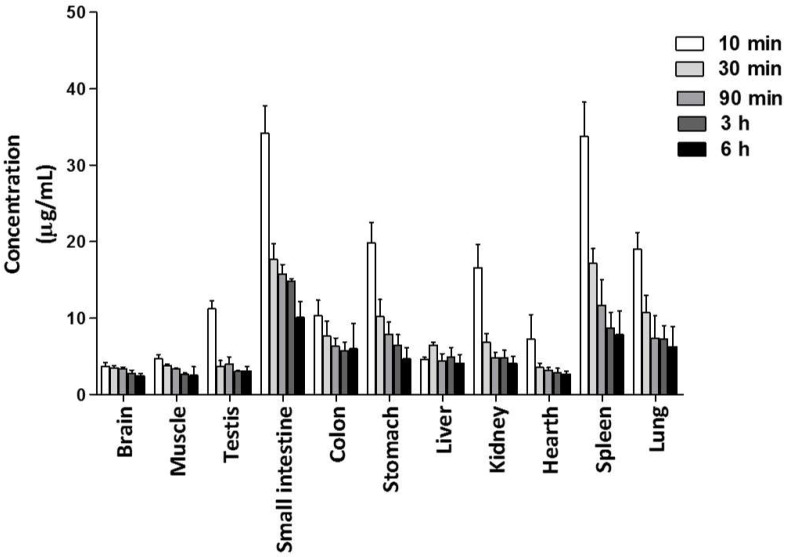
Concentration of **C1** in the distinct tissues of rats, evaluated at 10, 30, 60, 90, 180, and 360 min after administering a single p.o. dose (100 mg/dL) of the compound (mean ± SD, *n* = 3).

**Table 1 molecules-26-06801-t001:** Data on the intra- and inter-day precision and accuracy of the HPLC method for the quantification of **C1** in rat plasma. The intra- and inter-day accuracy and precision were determined using different quality control samples (*n* = 5) of **C1** at distinct concentrations (0.1, 10, and 100 µg/mL) on five consecutive days. Accuracy and precision were expressed in terms of relative error (RE) and relative standard deviation (RSD), respectively.

Nominal Concentration (µg/mL)	Measured Concentration (µg/mL) Expressed as the Mean ± SD	Accuracy (%)	Precision (C.V., %)
**Intra-day**			
0.1	0.09 ± 0.01	89.3	14.9
10	9.0 ± 0.4	89.5	4.4
100	99.5 ± 6.6	99.5	6.6
**Inter-day**			
0.1	0.09 ± 0.01	89.1	11.1
10	8.9 ± 0.5	81.5	5.9
100	99.3 ± 3.7	94.9	3.7

**Table 2 molecules-26-06801-t002:** The absolute recovery of **C1** from rat plasma with RP-HPLC. The recovery of **C1** in rat plasma, tissues and organs was carried out by comparing the average of the maximum areas of the quality control standard with those of the samples containing the standard solutions added to the matrix (plasma, tissues and organs), which had the same final concentration as the pure standards. The recovery of **C1** was assayed in triplicate at concentrations of 0.1, 10, and 100 µg/mL. The mean ± standard deviation (SD) of the percentage of absolute recovery is reported. The coefficient of variation (C.V., %) was determined for each of the evaluated concentrations of **C1**.

Nominal Concentration (µg/mL)	Recovery (%), Expressed as the Mean ± SD	C.V. (%)
0.1	89.3 ± 0.0	14.9
10	90.5 ± 0.4	4.4
100	99.5 ± 6.6	6.6

**Table 3 molecules-26-06801-t003:** The stability of **C1** in rat plasma. Quality control samples of rat plasma were used at different concentrations (0.1, 0.5, 10, and 100 µg/mL) in triplicate. The stability of **C1** was evaluated under various conditions: condition 1, at room temperature (25 °C) for 12, 24, and 48 h; condition 2, refrigeration (4 °C) for 24, 48, and 72 h; condition 3, three freeze–thaw cycles (from −20 to +25 °C) for 24, 48, and 72 h.

**Condition 1, Room Temperature (25 °C)**						
**Concentration** **(µg/mL)**	**Difference at 12 h (%) ***	**C.V.** **(%)**	**Difference at 24 h (%) ***	**C.V.** **(%)**	**Difference at 48 h (%) ***	**C.V.** **(%)**
0.1	10.7	14.9	3.2	14.5	3.8	8.9
0.5	3.2	11.9	7.8	9.3	4.1	6.2
10	9.5	4.4	6.5	8.8	3.7	4.0
100	0.5	6.6	6.5	7.2	5.4	1.5
**Condition 2, Refrigeration (4 °C)**						
**Concentration** **(µg/mL)**	**Difference at 12 h (%) ***	**C.V.** **(%)**	**Difference at 24 h (%) ***	**C.V.** **(%)**	**Difference at 48 h (%) ***	**C.V.** **(%)**
0.1	5.1	13.4	14.0	11.3	4.3	6.8
0.5	4.0	6.7	3.6	7.2	5.3	8.9
10	10.1	7.9	13.1	5.3	9.3	6.4
100	1.2	5.3	0.2	6.3	5.2	2.3
**Condition 3, Three Freeze–Thaw Cycles (From −20 to +25 °C)**						
**Concentration** **(µg/mL)**	**Difference at 12 h (%) ***	**C.V.** **(%)**	**Difference at 24 h (%) ***	**C.V.** **(%)**	**Difference at 48 h (%) ***	**C.V.** **(%)**
0.1	5.1	5.3	1.76	14.2	3.4	12.7
0.5	0.1	7.9	0.3	11.5	1.0	2.2
10	0.7	6.7	1.2	2.6	3.2	1.9
100	0.1	13.4	0.6	1.6	2.0	1.0

* Percent of difference in relation to the nominal concentration.

**Table 4 molecules-26-06801-t004:** Plasma pharmacokinetic parameters of **C1** in rats, determined by non-compartmental model analysis after a single i.v., i.p., and p.o. dose of 50, 75, or 50 mg/kg, respectively. Data are expressed as the mean ± SD (*n* = 6). Data are expressed as the mean ± SD (*n* = 6).

Parameters *	Units	p.o.	i.p.	i.v.
K_a_	h^−1^	0.9 ± 0.1	1.4 ± 0.2	-
C_max_	µg/mL	2.8 ± 0.1	22.0 ± 1.0	-
t_max_	min	33 ± 3	33 ± 5	-
Cp^0^	µg/mL	-	-	11.0 ± 1.9
K_e_	h^−1^	0.4 ± 0.1	0.5 ± 0.1	0.8 ± 0.3
Vd	L/kg	-	-	5.2 ± 0.9
CL	mL/min	-	-	24 ± 5
t_1/2a_	h	1.4 ± 0.3	1.3 ± 0.5	-
t_1/2e_	h	2.5 ± 0.6	1.4 ± 0.2	0.9 ± 0.2
MRT	h	1.7 ± 0.2	3.1 ± 0.9	1.7 ± 0.4
AUC_tot_	µg min^−1^/mL	618 ± 63	2780 ± 69	806 ± 72

* K_a_, absorption rate constant; C_max_, maximum plasma concentration; t_max_, time to maximum plasma concentration; Cp^0^, initial plasma concentration; K_e_, elimination rate constant; Vd, volume of distribution; CL, clearance; t_1/2a_, absorption half-life; t_1/2e_, elimination half-life; MRT, mean residence time; AUC_tot_, area under the plasma concentration-time curve from time zero to in-finity; F% p.o., fraction absorbed (absolute bioavailability).

**Table 5 molecules-26-06801-t005:** The percentage of the bound and unbound fractions of **C1** in rat plasma (± SD). The plasma protein binding assay was carried out by RP-HPLC with the ultrafiltration method at a concentration of 1, 5, 10, and 20 µg/mL of **C1** in the plasma of Wistar rats.

Concentration (µg/mL)	Percentage of Unbound Fraction ± SD (%)	Percentage of Bound Fraction ± SD (%)
1	96.3 ± 1.4	3.7 ± 1.1
5	94.2 ± 2.4	5.8 ± 2.4
10	93.4 ± 1.5	6.6 ± 1.5
20	94.8 ± 2.3	5.2 ± 2.3

**Table 6 molecules-26-06801-t006:** Blood/plasma (BP) partitioning allowed for the determination of the BP ratio. Samples of **C1** were prepared in the whole blood of rats at concentrations of 5 and 10 µg/mL (*n* = 3) and incubated at 37 °C for 4 h. Plasma was drawn from blood samples and the **C1** concentration was established using the RP-HPLC method. The BP ratio was calculated by dividing 5 or 10 µg/mL by the corresponding concentration in plasma separated from blood samples. Data are expressed as the BP ratio ± SD.

Concentration (µg/mL)	BP Ratio ± SD
5	0.54 ± 0.02
10	0.40 ± 0.01

## Data Availability

All the relevant data found in the study are available in the article. The data supporting the study are in Appendix A.

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
