# Peer review of "Preclinical Pharmacokinetics and Acute Toxicity in Rats of 5-{[(2E)-3-Bromo-3-carboxyprop-2-enoyl]amino}-2-hydroxybenzoic Acid: A Novel 5-Aminosalicylic Acid Derivative with Potent Anti-Inflammatory Activity"

_molecules, 2021, doi:10.3390/molecules26226801_

Round 1
Reviewer 1 Report
The Manuscript describes investigation of pharmacokinetics and toxicity in rats using one compound - 5-[(2E)-3-Bromo-3-carboxyprop-2-enoyl]amino}-2-hydroxybenzoic acid. The presentation of the results is good and I am convinced that the manuscript should be published. However, I have doubts whether the Molecules is a good choice.
The presentation of SI (non-published material) should be improved.
Author Response
Response to Reviewer 1 Comments
Point 1: The Manuscript describes investigation of pharmacokinetics and toxicity in rats using one compound -5-[(2E)-3-Bromo-3-carboxyprop-2-enoyl]amino}-2-hydroxybenzoic acid. The presentation of the results is good and I am convinced that the manuscript should be published. However, I have doubts whether the Molecules is a good choice.
Response 1: We appreciate the reviewer's comments. In this regard, in the Introduction section (page 2, lines 64-66), we have included the current need for alternative drugs for the treatment of diseases that involve inflammatory processes and capable of inhibiting a key enzyme in the process of inflammation which is myeloperoxidase (MPO), also that they do not generate adverse effects as some other drugs do.
Hence, we are convinced that 5-aminosalicylic acid (5-ASA) derivatives tested in the present work, could be promising compounds in the future for the treatment of diseases that involve inflammatory processes. Furthermore, it has been mentioned in the Introduction section (page 2, lines 70-77), that particularly C1 additionally to possess antioxidant properties in vitro, it has shown anti-inflammatory activity in an ear edema mouse model and can act as inhibitor of MPO, proving to exert an effect comparable to that of indomethacin [1]. In this regard, we have a registered patent for C1 [2].
Furthermore, the development of the present work has allowed us to conclude that C1 is not toxic to rats and that its pharmacnokinetic properties are adequate to be considered as a candidate drug. In fact, C1 shows some advantages over indomethacin and 5-ASA mentioned throughout the work. However, we know that it is necessary (as in the case for all the new molecules that are designed, synthesized, and proved) to continue with the preclinical evaluation of C1 and tested it in other models of diseases that involve inflammation, such as ulcerative colitis.
References
- Cabrera Pérez, L.C.; Gutiérrez Sánchez, M.; Mendieta Wejebe, J.E.; Hernández Rodríguez, M.; Fragoso Vázquez, M.J.; Salazar, J.R.; Correa Basurto, J.; Padilla Martínez, I.I.; Rosales Hernández, M.C. Novel 5-aminosalicylic derivatives as anti-inflammatories and myeloperoxidase inhibitors evaluated in silico, in vitro and ex vivo. Arabian J. Chem. 2019, 12, 5278-5291.
- Correa Basurto, J.; Rosales Hernández, M.C.; Mendieta Wejebe, J.E.; Trujillo Ferrara, J.G; Padilla Martínez, I.I.; Ramírez Durán, L.A.; Alemán González Duhart, D. Síntesis de un derivado del ácido 5-aminosalicílico con actividad antioxidante. Patente del Instituto Politécnico Nacional. Título de Patente No. 376240 y Solicitud No. MX/a/2013/007745. Instituto Mexicano de la Propiedad Industrial 2020. http://www.gob.mx/impi
Point 2: The presentation of SI (non-published material) should be improved.
Response 2: As it has been suggested by the reviewer, we have improved the presentation of the Supplementary Materials, making modifications in their global format, including the presentation of results, tables and graphs. This material is attached for your review.
Reviewer 2 Report
In the process of drug development, toxicity testing is indeed a very important step. If it can avoid side effected than others drug can achieve significant benefit on clinical drug therapy. There are this is a very interesting discovery.
Here has some simply questions
- What is the concentration and lethal dose of the C1 recent drug test? What is the difference in the PO IV for the converted animal dose?
- What is the result in anti-inflammatory effect to compare with the previous related drugs (such 5-ASA)? Could you please compare the half-life and toxicity test with these similar drugs? It will be help for the reader to understand the miner but important different.
- Here shown the median lethal dose (LD50) values in Wistar rats were >2000 and >1098 mg/kg for p.o. 90 and i.p. administration. The concentration is the Maximal Toxicity on rat? Or it is a lethal dose you did the test before? What is the sum of the concentrations, and what is the human dose?
4. Seemly, C1 has a relatively small effect on the gastrointestinal toxicity, but based on the current data, what the possible toxicity will be./
Author Response
Response to Reviewer 2 Comments
In the process of drug development, toxicity testing is indeed a very important step. If it can avoid side effected than others drug can achieve significant benefit on clinical drug therapy. There are this is a very interesting discovery.
Here has some simply questions
Point 1: What is the concentration and lethal dose of the C1 recent drug test?
Response: The LD50 value of C1 that we obtained in Wistar rats for the p.o. route of administration were >2000 mg/kg (page 3, line 93), which was calculated using the maximum likelihood method with the AOTStatPgm software indicated in the Guideline 425 (Acute Oral Toxicity: Up-and-Down Procedure) of the Organization for Economic Cooperation and Development [1]. According to this guideline, the maximum dose administered to animals can be 2000 mg/kg (using a maximum of 5 animals), and only under specific regulatory needs, a dose of up to 5000 mg/kg could be used. However, in the case of C1, the first animal was administered with a dose of 175 mg/kg followed by a dose of 550 mg/kg for the next animal, and finally a dose of 2000 mg/kg was administered to the rest of animals. Therefore, according to the Globally Harmonized System (GHS) of Classification and Labeling of Chemical Products, C1 is classified in the categories 5 for the p.o. route and 4 for the i.p. route, indicating a low risk of toxicity of this compound.
Point 1: What is the difference in the PO IV for the converted animal dose?
Response: In this study, we did not obtain the LD50 value for the i.v. route of administration, since we based on the Guideline 425 is indicated that the test procedure is valuable to minimize the number of animals and necessary to estimate the acute oral toxicity of new chemical compounds without any prior information on their toxicity [1]; however, this guideline is not indicated to evaluate the LD50 value by the i.v. route of administration. In this regard, it is important to mention that we decided from the beginning to evaluate the toxicity of C1 by the p.o. and i.p. routes of administration, since they are those that have an initial absorption process. However, it would be of great importance in the future to seek an adequate methodology to determine the LD50 value of C1 by the i.v. route of administration.
On the other hand, since in the present work we carried out the pharmacokinetic study of the new chemical entity named as C1, for which there is no information prior to this study of the dose administered in rats, we based on previous literature. Firstly, we chose the dose of 50 mg/kg to be administered for p.o. route, based on the pharmacokinetics of 5-ASA in its formulation as Sulfasalazine [2]. On the other hand, we chose the doses of 50 mg/kg and 75 mg/kg to be administered by i.v. and i.p. routes, respectively, based on the pharmacokinetics of 5-[(4-Carboxybutanoyl)Amino]-2-Hydroxybenzoic Acid, which is a derivative of 5-ASA [3].
We have included these references in the Material and Methods section to do this part more comprehensbile (page 13, lines 454-455).
Point 2: What is the result in anti-inflammatory effect to compare with the previous related drugs (such 5-ASA)?
Response: It has been reported previously that C1 shows anti-inflammatory activity, which was evaluated in a 12-O-tetradecanoylphorbol acetate (TPA)-induced mouse ear edema model, demonstrating to have an effect comparable to that of indomethacin, although in this study the anti-inflammatory effect of C1 was not compared with that of 5-ASA. However, in an in vitro model has been shown that C1 can reduce the production of the free radicals, 2,2´-azino-bis(3-ethylbenzo thiazoline)-6-sulfonic acid (ABTS) and 2,2´-diphenyl-1-picrylhydrazyl (DPPH), about 90% in comparison to 5-ASA which generates a lower reduction of this radical, being about 85% at the same concentration (0.408 mM) [4,5].
We summarize the aforementioned in the Introduction section (page 2, lines 70-77).
Point 2: Could you please compare the half-life and toxicity test with these similar drugs? It will be help for the reader to understand the miner but important different.
Response: As it has been suggested by the reviewer, the half-life value of C1 was compared with that of 5-ASA, being included in the Discussion section (page 10, lines 305-312).
Likewise, as indicated by the reviewer, in the Discussion section we have included, the comparison of the toxicity study between C1, 5-ASA and indomethacin (page 9, lines 262-273; page 10, lines 274-275).
Point 3: Here shown the median lethal dose (LD50) values in Wistar rats were >2000 and >1098 mg/kg for p.o. 90 and i.p. administration. The concentration is the Maximal Toxicity on rat?
Response: According to the Guideline 425 (Acute Oral Toxicity: Up-and-Down Procedure) of the Organization for Economic Cooperation and Development [1], which was the guideline we used for determining the LD50 of C1, the maximum dose administered to animals can be 2000 mg/kg (using a máximum of 5 animals), and only under specific regulatory needs, a dose of up to 5000 mg/kg could be used.
Point 3: Or it is a lethal dose you did the test before?
Response: There was no previous information about toxicity of C1 in animals, therefore in this work we describe the methodology to establish the acute toxicity of this compound in rats, which is mentioned in the Material and Methods section (page 11, lines 366-372; page 12, lines 376-374), as well as the respective results, which are mentioned in the Results section (page 3, lines 93-99).
Point 3: What is the sum of the concentrations, and what is the human dose?
Response: If we add the LD50 values of C1 obtained by the p.o (>2000 mg/kg) and i.p (1098 mg/kg) routes of administration, the LD50 value would be >3098 mg/kg. Hence, considering this, we have mentioned in the Results section (page 3, lines 96-99) that, according to the Globally Harmonized System (GHS) of Classification and Labeling of Chemical Products, C1 is classified in the categories 5 for the p.o. route and 4 for the i.p. route, indicating a low risk of toxicity of this compound.
Since C1 is a new compound, until now the LD50 in humans has not been determined. However, it has been reported that the LD50 value for 5-ASA is of 2800 mg/kg (Drug bank). Furthermore, it has been shown that an optimal dose of 5-ASA for inducing remission of ulcerative colitis is 0.5 g of this drug, 3 times a day [6].
Point 4: Seemly, C1 has a relatively small effect on the gastrointestinal toxicity, but based on the current data, what the possible toxicity will be./
Response: With the LD50 values that we obtained from the acute toxicity study of C1, we can affirm that it has a low risk of toxicity. However, it would be of great importance in the future that, based on the results obtained from pharmacokinetics in the present work, can be established a daily dosage regimen based on a therapeutic dose of C1, as well as to expand the studies of evaluation of the sub-acute and chronic toxicity of this compound. Also, it results necessary to determine the time that C1 takes to be eliminated from each of the tissues in which it is distributed, to know if it is not being deposited for long periods, and evaluate if it is not toxic in the long term with a repeated dosage regimen.
References
- Organization for Economic Cooperation and Development. OECD, Test No. 425: Acute Oral Toxicity: Up-and-Down Procedure. E.C.D. 2008, 1-27.
- Sjöquist, B.; Ahnfelt, N.; Andersson, S.; D'argy, R.; Fjellner, G.; Hatsuoka, M.; Ljungstedt-Påhlman, I. Pharmacokinetics of Salazosulfapyridine (Sulfasalazine, SASP). V. Pharmacokinetics of SASP after a single intravenous or oral administration in the dog. Drug Metab. Pharmacokinet. 1991, 6, 491-507.
- Romero-Castro, A.; Gutiérrez-Sánchez, M.; Correa-Basurto, J.; Rosales Hernández, M.C.; Padilla, Martínez, I.I.; Mendieta-Wejebe, J.E. Pharmacokinetics in Wistar Rats of 5-[(4-Carboxybutanoyl)Amino]-2-Hydroxybenzoic Acid: A Novel Synthetic Derivative of 5-Aminosalicylic Acid (5-ASA) with Possible Anti-Inflammatory Activity. PloS One 2016, 11, e0159889.
- Cabrera Pérez, L.C.; Gutiérrez Sánchez, M.; Mendieta Wejebe, J.E.; Hernández Rodríguez, M.; Fragoso Vázquez, M.J.; Salazar, J.R.; Correa Basurto, J.; Padilla Martínez, I.I.; Rosales Hernández, M.C. Novel 5-aminosalicylic derivatives as anti-inflammatories and myeloperoxidase inhibitors evaluated in silico, in vitro and ex vivo. Arabian J. Chem. 2019, 12, 5278-5291.
- Correa Basurto, J.; Rosales Hernández, M.C.; Mendieta Wejebe, J.E.; Trujillo Ferrara, J.G; Padilla Martínez, I.I.; Ramírez Durán, L.A.; Alemán González Duhart, D. Síntesis de un derivado del ácido 5-aminosalicílico con actividad antioxidante. Patente del Instituto Politécnico Nacional. Título de Patente No. 376240 y Solicitud No. MX/a/2013/007745. Instituto Mexicano de la Propiedad Industrial 2020. http://www.gob.mx/impi
- Kruis, W.; Bar-Meir, S.; Feher, J.; Mickisch, O.; Mlitz, H.; Faszczyk, M.; Chowers, Y.; Lengyele, G.; Kovacs, A.; Lakatos, L.; Stolte, M.; Vieth, M.; Greinwald, R. The optimal dose of 5-aminosalicylic acid in active ulcerative colitis: a dose-finding study with newly developed mesalamine. Gastroenterol. Hepatol. 2003, 1, 36-43.

Reviewer 3 Report
This manuscript evaluates the pharmacokinetics and acute toxicity in rats of compound C1, A novel 5-Aminosalicylic acid derivative that exhibits anti-inflammatory activity comparable to indomethacin. Before the determination of Pk and acute toxicity, the extraction method of C1 from plasm, homogenized tissues and organs and the concentration determination of C1 by HPLC method were validated. After that, LD50 and pharmacokinetics parameters were measured in three administration routes, intravenous (i.v.), oral (p.o.), and intraperitoneal (i.p.). In addition, distribution of C1 in tissues subsequent to p.o. administration, plasma protein binding and C1 blood/plasma ratio were also assessed. The above results demonstrate that C1 is not toxic to rats and C1 administered p.o. in rats is suitable for reaching the colon where it can exert its effect. However, there are serval mistakes and comments in the manuscript that needs to be addressed.
- In 2.3. Pharmacokinetic Profiles, Figure 3a, Figure 3b and Figure 3c should be Figure 4a, Figure 4b and Figure 4c
- Under Table 4, there is a Figure 77, should be Figure 7.
- The concentration of C1 in spleen was comparable to small intestine in figure 5. Could spleen also be one of the major absorption sites? Could authors explain the reason?
- Why the dose of 50mg/kg was used for i.v. and p.o. but dose of 75mg/kg for i.p. when measuring the PK parameters?
Author Response
Response to Reviewer 3 Comments
This manuscript evaluates the pharmacokinetics and acute toxicity in rats of compound C1, A novel 5-Aminosalicylic acid derivative that exhibits anti-inflammatory activity comparable to indomethacin. Before the determination of Pk and acute toxicity, the extraction method of C1 from plasm, homogenized tissues and organs and the concentration determination of C1 by HPLC method were validated. After that, LD and pharmacokinetics parameters were measured in three administration routes, intravenous (i.v.), oral (p.o.), and intraperitoneal (i.p.). In addition, distribution of C1 in tissues subsequent to p.o. administration, plasma protein binding and C1 blood/plasma ratio were also assessed. The above results demonstrate that C1 is not toxic torats and C1 administered p.o. in rats is suitable for reaching the colon where it can exert its effect. However, there are serval mistakes and comments in the manuscript that needs to be addressed.
Point 1: In 2.3. Pharmacokinetic Profiles, Figure 3a, Figure 3b and Figure 3c should be Figure 4a, Figure 4b and Figure 4c
Response: The figure number has been corrected in the Results section (page 6, lines 180, 184, and 186).
Point 2: Under Table 4, there is a Figure 77, should be Figure 7.
Response: We appreciate the obserrvation, however it is important to clarify that in the original version submitted, the mentioned text (Figure 77) was not appear under Table 4, but by any way, we have ensured that now either not appear (page 7, lines 198-201).
Point 3: The concentration of C1 in spleen was comparable to small intestine in figure 5. Could spleen also be one of the major absorption sites? Could authors explain the reason?
Response: Yes, in fact with the results obtained (Figure 5) we can say that the spleen is one of the main absorption sites for C1, which could be explained since this organ has particular structural characteristics that allow the molecules transported by the blood to be completely permeable by the splenic microcirculation [1, 2]. Based on this important property, it has been suggested that a large part of the drugs that are excluded from most other organs, is due to their properties of load, size, etc., thus can be distributed within the splenic parenchyma [1].
In this regard, we have included an explanation of this point in the Discussion section (page 10, lines 292-298).
Point 4: Why the dose of 50 mg/kg was used for i.v. and p.o. but dose of 75 mg/kg for i.p. when measuring the PK parameters?
Response: Since in the present work we carried out the pharmacokinetic study of the new chemical entity named as C1, for which there is no information prior to this study of the dose administered in rats, we based on previous literature. Firstly, we chose the dose of 50 mg/kg to be administered for p.o. route, based on the pharmacokinetics of 5-ASA in its formulation as Sulfasalazine [3]. On the other hand, we chose the doses of 50 mg/kg and 75 mg/kg to be administered by i.v. and i.p. routes, respectively, based on the pharmacokinetics of 5-[(4-Carboxybutanoyl)Amino]-2-Hydroxybenzoic Acid, which is a derivative of 5-ASA [4].
We have included these references in the Material and Methods section to do this part more comprehensbile (page 13, lines 454-455).
References
- Cataldi, M.; Vigliotti, C.; Mosca, T.; Cammarota, M.; Capone, D. Emerging Role of the Spleen in the Pharmacokinetics of Monoclonal Antibodies, Nanoparticles and Exosomes. Int. J. Mol. Sci. 2017, 18, 1249.
- Steiniger, B.; Bette, M.; Schwarzbach, H. The open microcirculation in human spleens: a three-dimensional approach. J. Histochem. Cytochem. 2011, 59, 639-648.
- Sjöquist, B.; Ahnfelt, N.; Andersson, S.; D'argy, R.; Fjellner, G.; Hatsuoka, M.; Ljungstedt-Påhlman, I. Pharmacokinetics of Salazosulfapyridine (Sulfasalazine, SASP). V. Pharmacokinetics of SASP after a single intravenous or oral administration in the dog. Drug Metab. Pharmacokinet. 1991, 6, 491-507.
- Romero-Castro, A.; Gutiérrez-Sánchez, M.; Correa-Basurto, J.; Rosales Hernández, M.C.; Padilla, Martínez, I.I.; Mendieta-Wejebe, J.E. Pharmacokinetics in Wistar Rats of 5-[(4-Carboxybutanoyl)Amino]-2-Hydroxybenzoic Acid: A Novel Synthetic Derivative of 5-Aminosalicylic Acid (5-ASA) with Possible Anti-Inflammatory Activity. PloS One 2016, 11, e0159889.
